# Physical, Chemical and Rheological Characterization of Tuber and Starch from *Ceiba aesculifolia* subsp. *parvifolia*

**DOI:** 10.3390/molecules26072097

**Published:** 2021-04-06

**Authors:** Lizette Suastegui-Baylón, Ricardo Salazar, Yanik I. Maldonado-Astudillo, Manuel O. Ramírez-Sucre, Gerónimo Arámbula-Villa, Verónica Flores-Casamayor, Javier Jiménez-Hernández

**Affiliations:** 1Facultad de Ciencias Químico Biológicas, Universidad Autónoma de Guerrero, Av. Lázaro Cárdenas s/n, Ciudad Universitaria Sur, Col. La Haciendita, Chilpancingo 39090, Guerrero, Mexico; suasteguibaylonlizette@gmail.com (L.S.-B.); yimaldonado@uagro.mx (Y.I.M.-A.); 2CONACyT, Universidad Autónoma de Guerrero, Av. Javier Méndez Aponte No. 1, Fracc. Servidor Agrario, Chilpancingo 39070, Guerrero, Mexico; rsalazarlo@conacyt.mx; 3Centro de Innovación, Competitividad y Sustentabilidad, Universidad Autónoma de Guerrero, Calle Pino s/n, Col. El Roble, Acapulco 39640, Guerrero, Mexico; 4Centro de Investigación y Asistencia en Tecnología y Diseño del Estado de Jalisco, A.C. Unidad Sureste, Carretera Sierra Papacal-Chuburná Puerto, Mérida 97302, Yucatán, Mexico; oramirez@ciatej.mx; 5Centro de Investigación y de Estudios Avanzados del Instituto Politécnico Nacional (CINVESTAV IPN), Unidad Querétaro, Libramiento Norponiente # 2000, Fraccionamiento Real de Juriquilla, Querétaro 76230, Querétaro, Mexico; garambula@cinvestav.mx (G.A.-V.); veroflores@cinvestav.mx (V.F.-C.)

**Keywords:** high purity starch, fiber, bioactive compounds, high crystallinity, low retrogradation

## Abstract

This work aimed to evaluate the physical, chemical and antioxidant properties of *Ceiba aesculifolia* subsp. *parvifolia* (CAP) tuber and determinate rheological, thermal, physicochemical and morphological properties of the starch extracted. The CAP tuber weight was 3.66 kg; the edible yield was 82.20%. The tuber presented a high hardness value (249 N). The content of carbohydrates (68.27%), crude fiber (15.61%) and ash (9.27%) from the isolated starch, reported in dry weight, were high. Phenolic compounds and flavonoid content of CAP tuber peel were almost 3-fold higher concerning the pulp. CAP tuber starch exhibited a pseudoplastic behavior and low viscosity at concentrations of 5–15%. Purity percentage and color parameters describe the isolated starch as high purity. Thermal characteristics indicated a higher degree of intermolecular association within the granule. Pasting properties describes starch with greater resistance to heat and shear. CAP tuber starch has X-ray diffraction patterns type A. The starch granules were observed as oval and diameters ranging from 5 to 30 µm. CAP tuber could be a good source of fiber and minerals, while its peel could be used for extracting bioactive compounds. Additionally, the starch separated from this tuber could be employed as a thickening agent in food systems requiring a low viscosity and subjected to high temperatures.

## 1. Introduction

Today, 85% of the world’s food is based on only about 20 domesticated species; nevertheless, the nutritional potential of wild species is enormous since many of them are nutritious, easy to obtain and have a pleasant taste [1]. Furthermore, they have a good amount of phytochemicals with functional properties, mainly phenolic compounds [2]. An important group within edible wild plants are roots (tuber, rhizomes and bulbs); these foods provide an important source of carbohydrates and are essential components in the local diet during seasonal food shortages [1].

*Ceiba aesculifolia* subsp. *parvifolia* is a tree belonging to the Bombacaceae family that has commercial and nutritional importance in some parts of central Mexico; the seeds, roots and flowers are commercialized as food; the thorns and bark are used to make crafts [3,4]. Medicinal properties for this species have been reported; the tree bark has antioxidant activity associated with the presence of phenolic compounds, such as terpenes, phenylpropanoids, isoflavones and coumarins [4,5]. The tubers of the CAP tree are edible and contain a large amount of water. They measure over 40 cm long and weigh between 4 and 5 kg [6,7]. To the best of our knowledge, up to the present time, the nutritional composition of this tuber is still little studied.

Studies aimed at assessing the nutritional composition and content of bioactive compounds in wild edible products allow identifying resources with nutritional potential and evaluate the possibility of nutritional adequacy of these foods in the diet [8,9].

Besides being a food resource, tubers are known to contain significant amounts of starch [10]. Starch is a biopolymer consisting of glucose units organized into two types of glucans: mainly linear amylose, and amylopectin, which is highly branched [11]. The functional characteristics of starch, mainly the rheological and gelatinization properties, define its use as a gelling agent, thickener, adhesive and film-forming agent in industries such as food, textiles and pharmaceuticals [12,13]. The composition and functional characteristics of each starch are highly dependent on its sources. It is, therefore, important to analyze the properties of starches obtained from different plant sources [11,14,15]; furthermore, due to the increase in the demand for foods caused by the population growth, it is important to look towards new sources of starch to replace the conventional ones, such as potatoes, corn and wheat [16].

This work aimed to evaluate the physical, chemical and functional properties of the CAP tuber. Furthermore, the starch isolated from this tuber was analyzed by rheological studies, differential scanning calorimetry (DSC), rapid viscoanalysis (RVA), X-ray diffractometry (XRD), Fourier-transform infrared (FT-IR) spectroscopy and scanning electron microscopy (SEM) and, thereby, provide information for the use of this resource.

## 2. Results and Discussion

### 2.1. Physical Characterization of Ceiba aesculifolia subsp. parvifolia Tuber

Table 1 presents results regarding the physical properties of CAP tuber. The mean tuber weight was 3.66 kg, which was a higher value than reported for other similar tubers, such as *Sechium edule* [17] and *Colocasia esculenta* [18]. The edible yield of the CAP tuber (82.20%) was similar to the value reported for chayote (*Sechium edule*) tuber, 86% [17], and it was higher than the edible yield of sagu (*Canna edulis*) rhizome, 68.54% [19]. The weight and percentage of edible yield of CAP tuber suggest that there is scope for its use. Color parameters of the tuber pulp (L* = 70.85, C* = 21.30, °h = 67.32) describe a luminous, reddish-yellowish and slightly saturated color. Despite its beginning, the pulp is white; when the peel is removed, enzymatic browning reactions induce the appearance of a reddish color due to exposure to oxygen and the contact between oxidoreductase enzymes and unsaturated compounds, mainly monophenols that lead to the synthesis of melanins [20].

The parameters derived from TPA curves for CAP tuber are presented in Table 2. The hardness (249 N) values were higher than previously reported for *Pachyrhizus ahipa* tuber (135.4 N) [21]. The highest hardness value in the CAP tuber may be linked to a greater quantity of fiber due to the influence of this component in wall thickness [22,23]; additionally, it has often been observed that cells of irregular shape, small size as well as a compact cell arrangement, result in greater hardness and cohesiveness [24].

### 2.2. Proximal Composition of Ceiba aesculifolia subsp. parvifolia Tuber Pulp

The results of the proximate composition determined for CAP tuber pulp are shown in Table 3. The total carbohydrate content calculated using the difference method was the major component (68.27%), followed by crude fiber (15.61%), ash (9.27%) and protein (3.64%). Lipids were minority compounds (3.18%). The high level of water content in the pulp was evident (88.34%). As in other plant foods, in CAP tuber, carbohydrates provide the highest caloric content with a minor contribution from lipid and protein. The content of fiber emphasizes its potential for developing new healthy food products. Fiber gives a feeling of satiety and improves intestinal functioning [25]. Moreover, the addition of fiber can result in the formulation of foods low in calories and with functional properties, such as water and oil holding capacity [26]. Ash content in CAP tuber pulp is higher than *Pachyrhizus erosus*, a root very similar to CAP tuber, that presented an ash content of 4.60% [8] and *Roscoea purpurea*, a wild tuber for which Misra et al. [27] reported content of 5.62%. The mineral intake is important to maintain good health. This makes the CAP tuber pulp a potential contributor towards a balanced diet [28].

### 2.3. Polyphenolic Content and Antioxidant Activity of Ceiba aesculifolia subsp. parvifolia Tuber

Phenolic compounds are the second most abundant organic molecules in plants, and their consumption is important by virtue of their antioxidant capacity and their ability to regulate cellular processes, such as the expression of some genes and protein phosphorylation [29], whereas flavonoids represent nearly 75% of the polyphenols consumed as part of the diet, the significance of the latter lies in their antioxidant, anti-inflammatory and anticarcinogenic properties [29,30].

Results, expressed in mg of gallic acid equivalents (GAE) and mg of catechine (CE) per gram of ethanolic extract (EE), showing total soluble phenols (TSP), total soluble flavonoids (TSF) and antioxidant activity (AA), are presented in Table 4. 

Peel exhibited a higher TSP and TSF content (377.99 mg GAE/g EE y 134.00 mg CE/g EE, respectively) than that of pulp (44.21 mg GAE/g EE y 17.84 mg CE/g EE, respectively). Traditional use of CAP bark has been reported for the treatment of tumors, inflammation and infections [7,31]. Furthermore, antibacterial, antifungal and anti-inflammatory properties of this have been tested [32], properties which might be attributed to phenolics and related bioactive compounds.

Owing to the complex reactivity of bioactive compounds, to assess the antioxidant capacities of food or extracts, at least two different assays must be employed [33]. In the present study, the in vitro AA of EE of pulp and peel of CAP tuber was determined by the DPPH and ABTS methods; as in the case of phenolic compounds, the smallest value of EC_50_, which corresponds to a higher antioxidant capacity, was found in the peel (1.22 µg/mL and 0.17 µg/mL for DPPH and ABTS, respectively) than in pulp (12.70 µg/mL and 2.47 µg/mL, respectively). The EC_50_ value in pulp was higher than those reported by Park et al. [34] for yacon (*Smallanthus sonchifolius*) tuber extract, with an EC_50_ value of 675.28 µg/mL by DPPH method. TSP and TSF content of CAP tuber peel was almost 3-fold higher concerning the pulp. This ties in with the strongest antioxidant potential. The texture of this structure is very hard and cannot be eaten; however, it could potentially be used for therapeutic purposes.

### 2.4. Characterization of Ceiba aesculifolia subsp. parvifolia Starch

#### 2.4.1. Color Parameters and Extraction Yield

Color is an important attribute to designate starch quality [35]. Table 5 illustrate the color parameters of CAP tuber starch. The high L* value (91.28) and lower a* (−0.06) and b* (2.45) values (indicating a low degree of redness and yellowness) evidence the high purity of the starch [36,37]. 

Similar results were reported by Boundries et al. [37] for white sorghum starch (*Sorghum bicolor*) with values of 92.91, −0.26 and 3.58 for L*, a* and b*, respectively; similarly, the starch of *Canna edulis* rhizome, exhibited and L* value of 94.38 and a* and b* values of −0.06 and 1.47, respectively [38]. The color values reported for the isolated starch in the present work underline its potential for industrial applications in products where clearness is required.

The extraction yield of CAP tuber starch was 23.86%; this is a relatively low value when compared with other tubers, such as Solanum tuberosum (32.1%) [39] and Colocasia esculenta (81%) [40]. Nonetheless, a single root of CAP tree may produce at least four tubers, and it is deemed that a fully grown tree will have more than 100 tubers, just in their surface roots [41]. Although starch is not stored in large quantities in this tuber, the large biomass production of this tree makes it a viable source for starch extraction.

#### 2.4.2. Chemical Composition

The proximate analysis of CAP tuber starch was performed, and the results are showed in Table 6. The moisture content (10.43%) was lower than those reported for *Ensete ventricosum* (14%), *Solanum tuberosum* (13.5%) and *Zea mays* (13%) starches by Gebre-Mariam et al. [12]. Lipid content in CAP tuber starch (0.37%) was higher than maize starch (0.10%) [42] and starches from six potato varieties (0.15–0.34%) [43], lipid content importance lies in their nutritional implications, since complex amylose-lipid could produce resistant starch [44]. The proteins associated with the starch are of two types, storage proteins, which stayed on the surface of granules after extraction, and proteins granule-associated, which are strongly attached to integral components or surface of the starch granules. The quantity of protein that lingered in the starch after extraction procedures are related to its botanical source and the affinity of proteins for the granule surface [45]. Proteins were the component detected in larger quantities (1.07%) in CAP tuber starch; this value is higher than contents reported for other tuber starches, such as Solanum tuberosum [12], Oxalis tuberosa [42] and ñame varieties (*Dioscorea* spp.) [46], which presented values of 0.20, 0.34 and 0.40–0.60%, respectively. However, it was within the range reported for Andean tuber starches, 1.13–1.18% [43] and Bambusa vulgaris culm starch, which present a protein content between 1.27 and 2.10% [47]. The technological importance of minor components (proteins and lipids) presents in starch has been considered to be important in maintaining granule integrity acting as a protective coating against enzyme attack and disruption [45]. Ash was the component identified in a lower proportion (0.28%). The purity of CAP tuber starch (>98%) indicates that the starch isolation method results in starch with a low amount of impurities; furthermore, it may be considered as a high purity starch [48].

#### 2.4.3. Rheological Measurements

The rheological characterization of starches provides information about their structure and behavior under processing conditions, being useful for guiding their applications [49]. When rheological tests are carrying out within linear viscoelastic region, three important parameters are obtained: *G′*, which refers to the material elastic character, elastic energy can be stored and subsequently recovered; *G″,* associated with the viscous character, viscous energy is dissipated and lost as heat; and tan(*δ)=*(*G″/G′*), which relates the previous two parameters and is calculated as the ratio of the strain energy that is stored and lost [50]. The viscoelastic properties of CAP tuber starch and corn starch (used as control) are presented in Table 7. At all concentrations of CAP tuber starch, the *G′* values were higher than *G″* values; these describe an elastic behavior; on the other hand, tan *δ* values were bigger than 0.1 ranged from 0.13 to 0.16, this describes the gel formed by this starch as a weak gel since they are not able to store the applied energy and is lost as heat. By comparison, corn starch gel showed strong gel behavior (tan(*δ)* = 0.03). *η** is a measure of the overall resistance to flow and shows the evolution of the viscoelastic behavior of materials subjected to oscillatory shear [51,52]. This parameter showed a proportional increase to CAP tuber starch concentration; nevertheless, the values were lower than those shown by corn starch.

To simplify the comparison of flow curves obtained in rheological tests, mathematical models that adjust the numerous measurement points are used, resulting in a small number of parameters that are easily comparable [51] The power-law model (Table 8) was satisfactorily (R^2^ = 0.99) used to evaluate the flow behavior of CAP tuber starch gel. Shear-thinning behavior shows an *n* < 1, whereas an *n* > 1 describes a shear-thickening behavior [53]. At a concentration of 10% and 15% (*w*/*v)*, CAP tuber starch exhibit a shear-thinning behavior (*n* = 0.30, 0.38, respectively), as well as corn starch at 10% (*n* = 0.16). *K*, a parameter that corresponds to the gel viscosity, had a much higher value (122.771 Pa·s) for corn starch than for CAP tuber starch (33.799 Pa·s).

The viscosity patterns of CAP tuber and corn starches are shown in Figure 1; both behaved like a non-Newtonian fluid since viscosity fluctuates with shear rate [54], as well it is possible to observe that viscosity decreases as shear rate increases, a characteristic pattern of a shear-thinning behavior [55]. CAP tuber starch develops a non-Newtonian and shear-thickening gel, similar to other tubers starch; however, it develops a weak gel with a low viscosity. Therefore, it could be employed in foods where low viscosity is needed, such as soups, sauces, fruit nectars and dressings; nevertheless, its applications will depend on the concentration of hydrocolloid and the food system in which it will be used; inasmuch as, synergistic relationships between ingredients and exposure temperatures during manufacture have been observed [56].

#### 2.4.4. Thermal Properties

DSC is the most used technique to study the starch gelatinization process. During this procedure, heat flow, which is generated as a result of endothermal and exothermic events, is monitored in the sample [57]. Figure 2 shows the thermogram obtained during the DSC analysis of the starch and revealed an endothermic peak related to starch gelatinization at 79 °C. The gelatinization temperatures are shown in Table 9. The values of onset temperature (To), peak temperature (Tp) and conclusion temperature (Tc) of CAP tuber starch were higher than those found for starch from other botanical sources, such as *Solanum tuberosum* (58.7, 62.6 y 68.1 °C, respectively), *Zea mays* (63.2, 69.0 y 75.2 °C, respectively) and *Ensete ventricosum* (61.8, 65.2 y 71.7 °C, respectively) [12]. The gelatinization temperatures are strongly related to the thermal stability of the crystalline structure and reflect the degree of intragranular ordering. CAP tuber starch presents more crystalline regions that are thermally and structurally stable due to its high gelatinization temperatures [58,59]. Thermal stability of CAP tuber starch would be a desirable feature for the development of food products that are subjected to high temperatures, such as canned and baby food [60]. Furthermore, CAP tuber starch gelatinization enthalpy (ΔH) was lower (16.82 J·g^−1^) than reported for *Solanum tuberosum* (19.8 J·g^−1^) and *Ensete ventricosum* (21.6 J·g^−1^), in the above-mentioned study, and was within the range of reported values for *Colocassia esculenta* starch (15.8 an 18.2 J·g^−1^) by Lu et al. [61]. ΔH value gives an overall measure of the energy needed to dissociate the internal structure of starch granules and is proportional to the degree of molecular compaction within the granule [13,62]. The lower ΔH obtained for CAP tuber starch could be related to its shorter particle size due to shorter amylopectin chains, demanding lower energy to cleave these structures [40].

#### 2.4.5. Pasting Properties

The pasting properties of CAP tuber starch are presented in Figure 3 and Table 9. Pasting temperature (P_temp_) represents the temperature at which viscosity starts increasing, the value obtained for the starch of CAP tuber was higher (79.15 °C) than those determined for three potato cultivars starches, ranged from 70.0 to 73.5 °C [63], *Sechium edule*, 67.75 °C and *Zea mays*, 78.55 °C [17]. On the other hand, peak viscosity (Pv), final viscosity (Fv), breakdown viscosity (BDv), and setback viscosity (SBv) determined in CAP tuber starch were all lower than those reported for starch of three potato cultivars (3196 to 3467 cP, 2564 to 3111 cP, 1037 to 1460 cP and 520 to 397 cP, respectively), *Sechium edule* (14746 cP, 4939 cP, 12417 cP and 2610 cP, respectively) and *Zea mays* (4959 cP, 4237 cP, 1724 cP and 1037 cP, respectively) in the above-mentioned works. Higher P_temp_ and lower Pv of CAP tuber starch may be related to the presence of minor components, such as lipids, sugars and proteins [64,65]. High BDv values describe a starch that breaks down easily [66]; hence, the lower BDv observed in CAP tuber starch suggested its greater resistance to heat and shear than other starches; this less fragility can be related to the stability of the crystalline structure inside the granule. SBv refers to the tendency of starch pastes to retrograde due to the rearrangement of amylose molecules during cooling [67]. The lower retrogradation tendency of CAP tuber starch may be due to a low amylose content and the predominance of short-chain amylopectin [68]. From the obtained results, it could be argued that CAP tuber starch manifested thermal properties comparable and even better than other starches, such as potato and corn, although it develops lower viscosity.

#### 2.4.6. X-ray Diffractometry

X-ray diffractometry allows characterizing the crystalline structure of starch granules; the diffraction pattern can be considered as the fingerprint of each starch [69,70]. The different diffraction patterns are associated with different amylopectin branch chain length; type A consists mainly of short chains (23–29 units); type B has mostly long chains (30–44 units), and type C is made up of intermediate-sized chains (26–29 units). The X-ray diffraction pattern of CAP tuber starch presented in Figure 4 displays peaks at 15°, 18° y 23°, typical of starch type A; pattern attributed to short chains amylopectin. This is in line with the fact that CAP tuber starch showed ΔH values lower than other starches. It has been proposed that differences between diffraction patterns are due to amylopectin chains packing and the water content within the structure, type A starches consist mainly of compact short chains of amylopectin with a low water content [71,72]; as a result, this type of starches keep their crystalline structure when subjected to temperature fluctuations inasmuch as are less sensitive to these [73], this is consistent with the highest Tp determined in CAP tuber starch in the present study.

#### 2.4.7. Fourier Transform Infrared (FT-IR) Spectroscopy

FT-IR spectra provide information on a short-range order, close to the granule surface, about double-helical order and complement the data obtained through X-ray diffraction, which provides information about long-range order related to the packing of double helices [74,75]. Figure 5 depicts the FT-IR spectra corresponding to CAP tuber starch. Starch granules consist of two different water types, according to their location; one of these refers to water as a part of the hydrogen bonding network within the crystalline structure and another that is more mobile and considers hydration water [76]. Bands in the region 3000−3800 cm^−1^ may be attributed to O−H stretching regions of water molecules hydrogen-bonded [77], whereas the band observed around 2100 cm^−1^ is assigned to –OH bending vibration of free water molecules due to its hygroscopic nature [78]. In the region known as the fingerprint (1500–600 cm^−1^) [79], peaks discernible at 577 are assigned to skeletal modes of the pyranose ring [78]. The band at 930 cm^−1^ is interpreted as the skeletal mode of α 1→4 glycoside bonds [78,80]. Similarly, peaks observed at 1080 and 1158 cm^−1^ have been assigned to stretching vibration of C–O–H and C–O–C from glycosidic bonds [81]. Absorption at 1022 and 1047 cm^−1^ is related to amorphous and crystalline regions of the starch, respectively; the more pronounced the peak at 1047 cm^−1^ indicates a reduction in the amorphous area and an increase in the ordered region [74]. In CAP tuber starch, these bands were slightly distinguishable, although the band around 1047 cm^−1^ was sharper than that observed at 1022 cm^−1^, indicating a predominance of crystalline areas over the amorphous component. The band’s intensity at 1242 cm^−1^ is due to CH_2_OH and C–O–H deformation mode [80]. High pasting temperature and low viscosity determined for CAP tuber starch through RVA analysis in the present study suggests the presence of minor components. Absorption at 1638 cm^−1^ corresponded to lipids and proteins bounded to the carboxyl and carbonyl groups of starch [74]. In the same way, the presence of bands around 2930 and 2860 cm^−1^ could be attributed to protein attached to starch [82,83].

#### 2.4.8. Scanning Electron Microscopy

The individual morphological characteristics for each type of starch depend on plant physiology and the amount and types of enzymes involved in the biosynthetic process within the amyloplast, which in turn influences the swelling power and water-binding capacity of the granules [84]. The micrographs of the granules of CAP tuber starch are shown in Figure 6. In general, had an oval shape with diameters ranging from 5 to 30 µm, the surface appeared to be smooth, with any fissures; in some granules, a lump is observed at one end (Figure 6c). On the surface of some granules, laminar structures can be observed (Figure 6d), which were believed to be proteins and lipids [15,79]. CAP tuber starch presented an average diameter similar to those reported by Jane et al. [84] for sweet potato, water chestnut and jicama, all three with axes from 5 to 30 µm. In this same work, the granule size of the root and tubers starches analyzed ranges from 10–100 µm; in this sense, the CAP tuber starch granule can be considered as medium-low size.

## 3. Materials and Methods

### 3.1. Materials

Plant material of CAP was collected in San Juan Totolcintla, municipality of Mártir de Cuilapan, Guerrero, México (17°54′32″ N, 99°19′39″ W). The plant samples were deposited in the herbarium of the Universidad Autónoma de Guerrero and they were taxonomically identified. Later, CAP tubers were collected.

### 3.2. Physical Characterization of Ceiba aesculifolia subsp. parvifolia Tuber

Size, weight, color and texture were determined in CAP tubers. The size was evaluated measuring the longitudinal (LD) and transverse diameters (TD); the color of the peel and pulp were registered by employing a portable colorimeter Ci62 (X-Rite, Grand Rapids, MI, USA) as L* (brightness), C* (chromaticity) and °h (Hue angle) at different periods from the determined CIELCH. The illuminant employed was D65, and the standard observer position was 10°. The edible yield of CAP tubers was measured by dividing edible portions from non-food tissues and using the following formula:(1)$$\mathrm{Edible}\;\mathrm{yield}\;\left(\%\right)=\;\frac{\mathrm{Weight}\;\mathrm{of}\;\mathrm{edible}\;\mathrm{portion}\;(g)\;}{\mathrm{Total}\;\mathrm{weight}\;(g)}\;\times \;100$$

Texture measurement was made through a texture profile analysis (TPA) with a texture analyzer (TA-XT.plus, Stable Micro Systems, Surrey, UK). The tubers were washed, and the epicarp peeled off; the pulp was then cut into 1 cm cubes. Each sample was compressed with a 30 mm diameter cylindrical probe. The cross-head speed was 2 mm/s, and the maximum extent of deformation was 50% of the original length. The quality attributes measured were hardness (H), fracturability (F), adhesiveness (A), springiness (S), cohesiveness (C) and resilience (R). Results of size, weight, color and edible yield are presented as the average of four repetitions, while the results of TPA represent the average of ten repetitions.

### 3.3. Chemical Composition Analysis of Ceiba aesculifolia subsp. parvifolia Tuber

The nutrient content was calculated in CAP tuber pulp. Moisture, ash and crude fat content were determined according to AOAC international standard procedures 20.013, 923.03 and 920.39 [85], respectively. The crude protein (N × 6.25) was obtained by improved Kjeldahl method 46-16.01 of AACC [86]. The crude fiber was estimated by the 978.10 AOAC method [87]; finally, the total carbohydrate was assessed through difference. For the measurement of TSP, TSF and AA in the pulp and peel of CAP tuber, ethanolic extracts were prepared by macerating 10 g of fresh sample in 100 mL of ethanol, shaken manually for 10 min and then sonicated for 1 h at 40 °C in an ultrasonic cleaner (SB3200, Ningbo Scientz. Biotechnology Co., Ltd., Ningbo, China). The samples were centrifuged for 20 min at 10 °C and 3000 rpm. The resulting extracts were stored without light at 4 °C for 24 h.

The content of TSP was determined following the methodology of Singleton et al. [88]. TSF content was estimated as reported by Zhishen et al. [89]. The results were expressed as mg of gallic acid equivalents (GAE) and mg of catechine (CE) per gram of ethanolic extract (EE), respectively. AA in the ethanolic extracts was determined by the DPPH and ABTS^•+^ methods, both proposed by Thaipong et al. [90] with some modifications. Briefly, in the DPPH assay, a 1 mM stock solution was prepared in ethanol. The working solution was obtained by placing 10 mL of the stock solution in a volumetric flask and filling up to a volume of 100 mL with the same solvent to obtain a concentration of 0.1 mM. From a starting volume of 100 μL of the pulp and peel extracts, six serial dilutions were made in 50 μL of ethanol in a 96-well plate. The extracts were allowed to react with 150 μL of the DPPH solution for 30 min in the dark. The absorbance was taken at 517 nm in a spectrophotometer (Evolution 201, Thermo Fisher Scientific, Waltham, MA, USA). In the ABTS^•+^ assay, the modifications were from a starting volume of 100 μL of the pulp and peel extracts, and six serial dilutions were made in 50 μL of methanol in a 96-well plate. The extracts reacted with 150 μL of the ABTS^•+^ solution for 30 min in the dark. The inhibition of the DPPH and ABTS^•+^ radicals was calculated using Equation (1):(2)$$\mathrm{Inhibition}\;\left(\%\right)\;=\left[1\;-\;\left[\frac{\mathrm{Sample}\;\mathrm{absorbance}}{\mathrm{Control}\;\mathrm{absorbance}}\right]\right]\;\times \;100$$

The concentration of the extract that provided 50% free radical scavenging activity (EC_50_) was calculated from a graph of inhibition against the extracts’ concentration.

### 3.4. Characterization of Ceiba aesculifolia subsp. parvifolia Starch

#### 3.4.1. Starch Isolation

CAP tuber starch was isolated following methods described by Jiménez-Hernández et al. [17] with some modifications. Briefly, the tubers were washed, cut into small pieces, and placed in a 3% (*w*/*v*) citric acid solution for 10 min to prevent oxidation. Next, the pieces were homogenized in distilled water (1:3; pulp: water) in a blender. The resulting slurry was passed through fine muslin cloth and then filtered using a 250 μm-mesh sieve. The liquid fraction (starch) could settle for 24 h at 4 °C, and the residue (fiber) was subjected to the extraction process twice again. Later, starch-containing fractions were mixed and centrifuged at 3000 rpm for 10 min. The starch was dried in an air-circulating oven (DHG-9070A, Luzeren^®^, México City, Mexico) at 40 °C for 24 h.

#### 3.4.2. Color Analysis

Color of starch was determined using a portable colorimeter Ci62 (X-Rite, Grand Rapids, MI, USA), illuminant D65 and 10° standard observer position. L* to represent lightness (0 = black, 100 = white), a* (+a = redness, −a = greenness) and b* (+b = yellowness, −b = blueness) parameters were measured.

#### 3.4.3. Chemical Composition and Starch Yield

The moisture, ash and crude fat of isolated starch were determined according to AOAC (1990, 2005) international standard procedures (20.013, 923.03 and 920.39, respectively) [85]. Crude protein content (N × 6.25) was determined by improved Kjeldahl method 46-16.01 of AACC [86]. The purity percentage of isolated starch was determined as described by Cruz et al. [43] using Equation (2):
(3)$$\mathrm{Purity}=[100\%\;\left(\mathrm{dry}\;\mathrm{basis}\right)\;-\;\left(\mathrm{ash}\%\;+\;\mathrm{lipid}\%\;+\;\mathrm{protein}\%\right)]$$

The starch yield was obtained as described by Pascoal et al. [81].

#### 3.4.4. Rheological Studies

Starch gels at 5, 10 and 15% (*w*/*v*) were prepared as follows: weighed amounts of starch were mixed with distilled water and immersed in a boiling water bath using a magnetic stirrer equipped with a hot plate. Once a homogeneous dispersion was obtained, the temperature was raised until gelatinization point, which was about 85 °C, and then held at this temperature for 15 min to ensure full viscosity development. Rheological characteristics of starch gels were measured using oscillatory shear test following Tarrega et al. [91] with modifications. Starch gels were loaded onto a rotational rheometer (DHR-2, TA Instruments, New Castle, DE, USA) by employing a parallel geometry (plate SST ST 40 mm sandblast) with a 1000 μm gap. All determinations were carried out in duplicate at a temperature of 25 °C.

Viscoelastic properties were measured using oscillatory shear tests. To determine the linear viscoelastic region, stress sweeps were run at 1 Hz. After this, frequency sweeps of starch gels at 10 and 15% were performed using a frequency range of 0.1 to 100 rad/s, with the strain set at 0.1%, whereas in the starch gel at 5%, the frequency range was 0.01 to 100 rad/s and the strain set at 0.01%. Storage or elastic modulus (*G′*), loss or viscous modulus (*G″*) the loss tangent angle (tan(*δ)*) and the complex viscosity (*η**), as a function of frequency, were determined using the Trios v4.5.0.42498 Discovery HR TA instruments software.

Flow curves of the prepared starch gels were obtained at the range of shear rate between 0.01 and 1000 s^−1^, the data were fitted to the power-law (Equation (2)) model:(4)$${\sigma \;=\;K\gamma }^{n}$$
where σ (Pa) is the shear stress, K (Pa·s^n^) represents the consistency index, γ (s^−1^) is the value of shear rate, and n (dimensionless) is the flow behavior index.

#### 3.4.5. Differential Scanning Calorimetry (DSC)

The thermal properties of starch were determined with a differential scanning calorimeter (DSC 821e, Mettler-Toledo Inc., Schwerzenbach, Switzerland). The equipment was calibrated with indium, and an empty pan was used as a reference. Deionized water (10.0 mg) was added to 5.0 mg of starch in an aluminum pan that was sealed. After equilibration at room temperature for 1 h, samples were heated from 30 to 100 °C at 10 °C min^−1^. To, Tp, Tc and ΔH were determined by the Mettler-Toledo software.

#### 3.4.6. Pasting Properties

Experiments were performed according to AACC 61-02 method [92], using a rapid visco analyzer (3D, Newport Scientific Pty. Ltd., Warriewood, NSW, Australia). Parameters recorded were P_temp_, Pv, Tv, Fv, BDv and SBv and were expressed in centipoise (cP).

#### 3.4.7. X-ray Diffractometry (XRD)

The starch’s crystalline structure was obtained on an X-ray diffractometer (DMAX-2100, Rigaku, Tokyo, Japan), operated at 30 kV and 16 mA. The X-ray source was Cu Kα radiation with a wavelength of λ = 1.5405 Å. The sample was placed on a glass surface and scanned from 2θ = 5 to 50° (θ standing for the angle of diffraction).

#### 3.4.8. Fourier Transform Infrared (FT-IR) Spectroscopy

The FT-IR spectrum of starch was obtained using an infrared spectrometer (Spectrum GX, PerkinElmer, Waltham, MA, USA), equipped with a diffuse reflectance system. A region from 400 to 4000 cm^−1^ was used for scanning at a resolution of 4 cm^−1^ for 24 scans.

#### 3.4.9. Scanning Electron Microscopy (SEM)

The starch’s morphological features were examined with an environmental scanning electron microscope (ESEM XL30, Philips, Eindhoven, The Netherlands). The starch was spread on a stainless-steel plate and coated with graphite by a sputter-coater (JFC-1100, JEOL Ltd., Tokyo, Japan). The starch samples were placed in the SEM chamber and examined with a beam of 10–20 kV and using a gaseous secondary electron detector. The morphology of the granules was examined at 150×, 500×, 1000× and 1500×, 1 torr with a spot size of 4.6.

### 3.5. Statistical Analysis

Data reported were an average of triplicate observations and were expressed as mean ± standard deviation. Results of tests for compounds and antioxidant activity were analyzed by Student’s t-test (*p* < 0.05) using the statistical program SPSS 20.0.

## 4. Conclusions

CAP tuber has good dietary potential; its size, ash and lipids contents were higher than other wild and cultivated tubers; further, it contains a considerable amount of fiber. Tuber peel demonstrated a higher content of antioxidant compounds and antioxidant capacity than the pulp. Therefore, the pulp could be used to develop new foods with functional properties, while the peel could be put toward the extraction of bioactive compounds.

The percentage of purity indicates that the starch isolation method results in starch with a low amount of impurities, and it may be considered as a high purity starch. This assertion is reinforced by the high lightness (L*) and lower a* and b* values.

The gelatinized CAP tuber starch showed a non-Newtonian and shear-thinning flow behavior; moreover, it behaves as a weak gel with a low viscosity. Physicochemical characterization methods, such as paste properties and FT-IR, reveal the presence of minor components (proteins and lipids) in CAP tuber starch, a feature that can influence the development of low viscosity. XRD patterns showed an A-type crystal structure; for this reason, they exhibited higher resistance and heat stability. This, in turn, relates to its lower melting enthalpy and higher gelatinization temperatures determined by DSC. These results corroborate that CAP tuber starch could be used to control consistency in food systems where low viscosity is required and in foodstuffs subjected to high temperatures, like dairy products, canned goods, and baby food. In addition, pasting properties results revealed the lower retrogradation tendency and greater resistance to heat and shear of this starch. While extraction yield is barely acceptable, the great quantity of tubers produced per plant increases their potential for use. These all properties make CAP tuber starch a good option as a new functional ingredient for the food industry.

## Figures and Tables

**Figure 1 molecules-26-02097-f001:**
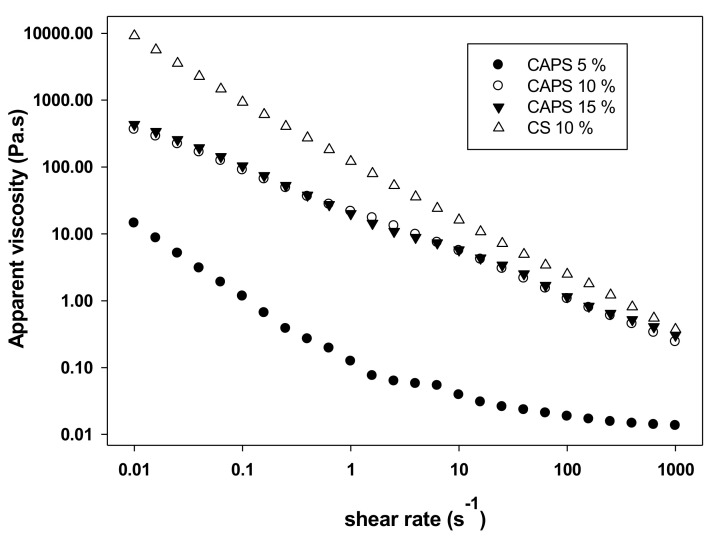
Steady shear viscosity of *Ceiba aesculifolia* subsp. *parvifolia* tuber starch (CAPS) and corn starch (CS).

**Figure 2 molecules-26-02097-f002:**
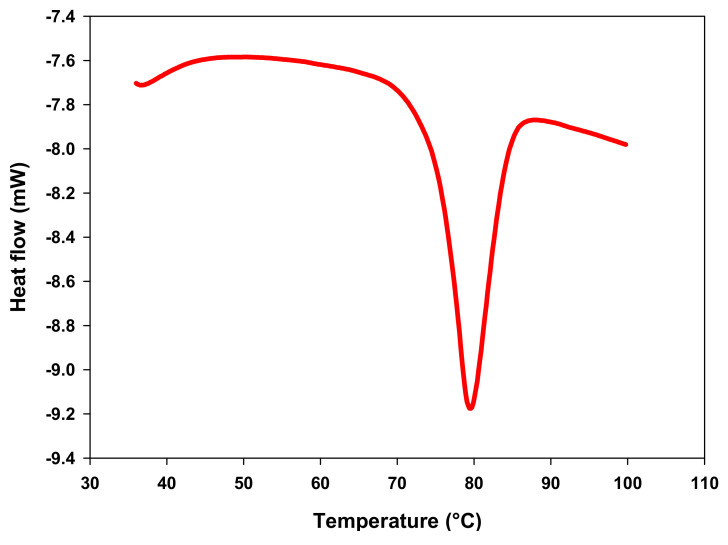
Differential scanning calorimetry (DSC) thermogram of *Ceiba aesculifolia* subsp. *parvifolia* tuber starch.

**Figure 3 molecules-26-02097-f003:**
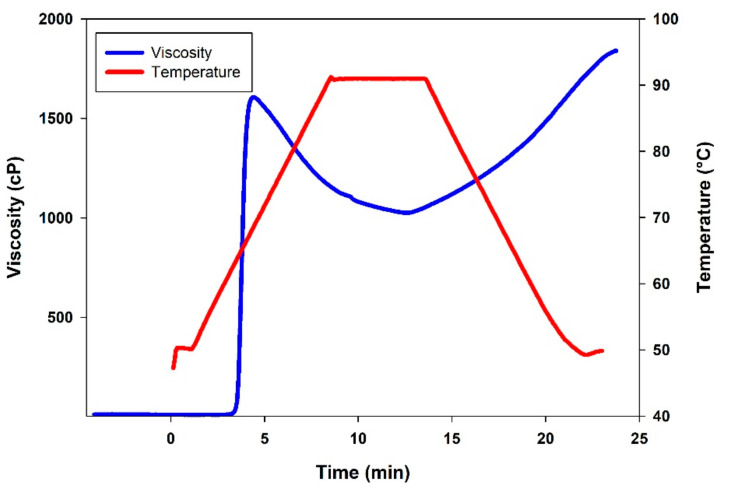
Rapid viscoanalysis (RVA) pasting curve of *Ceiba aesculifolia* subsp. *parvifolia* tuber starch.

**Figure 4 molecules-26-02097-f004:**
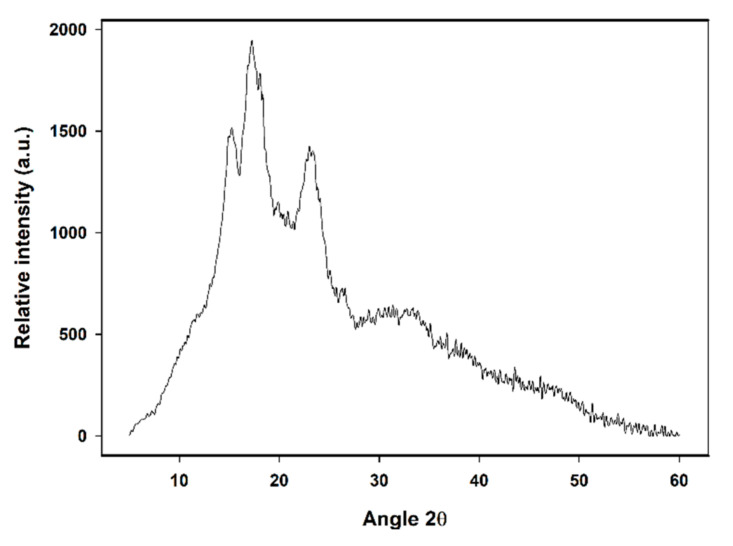
X-ray diffractogram of *Ceiba aesculifolia* subsp. *parvifolia* tuber starch.

**Figure 5 molecules-26-02097-f005:**
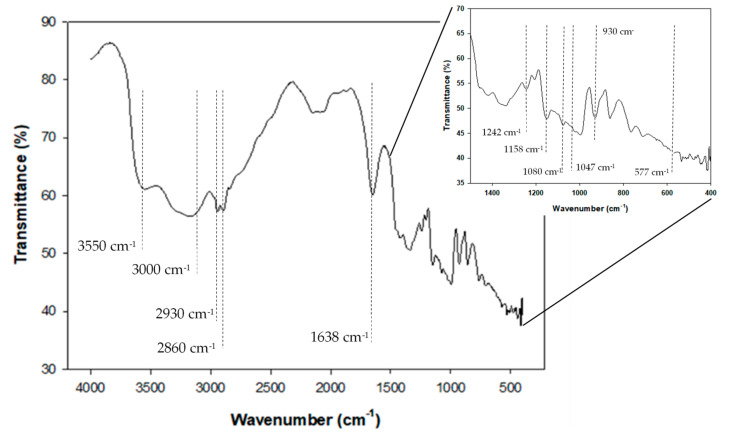
FT-IR spectra of *Ceiba aesculifolia* subsp. *parvifolia* tuber starch.

**Figure 6 molecules-26-02097-f006:**
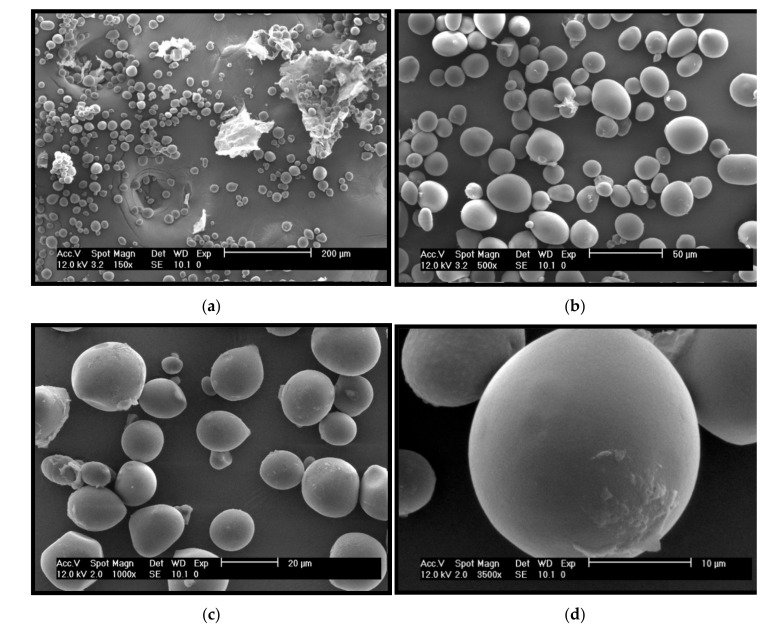
Scanning electron micrographs of *Ceiba aesculifolia* subsp. *parvifolia* tuber starch: (**a**) 150×; (**b**) 500×; (**c**) 1000×; (**d**) 3500×.

**Table 1 molecules-26-02097-t001:** Physical characterization of *Ceiba aesculifolia* subsp. *parvifolia* tuber.

Parameter		Mean Value ± SD
Weight (kg)		3.66 ± 1321
Longitudinal diameter (cm)		48.00 ± 14.49
Transverse diameter (cm)		42.50 ± 7.64
Edible yield (%)		82.20 ± 4.74
Pulp color	L*	70.85 ± 11.32
	C*	21.30 ± 5.58
	°h	67.32 ± 8.03
Peel color	L*	33.85 ± 3.36
	C*	10.74 ± 1.56
	°h	58.59 ± 2.93

Data are expressed as means ± standard deviation of fourfold experiments. L* (brightness), C* (chromaticity) and °h (Hue angle).

**Table 2 molecules-26-02097-t002:** Texture profile analysis of *Ceiba aesculifolia* subsp. *parvifolia* tuber pulp.

Parameter	Mean Value ± SD
Hardness (N)	249.08 ± 44.03
Fracturability (N)	138.84 ± 32.57
Adhesiveness (N)	0.022 ± 0.008
Springiness	0.76 ± 0.02
Cohesiveness	0.47 ± 0.08
Resilience	0.25 ± 0.06

Results are the means of ten determinations ± standard deviation.

**Table 3 molecules-26-02097-t003:** Proximate composition of *Ceiba aesculifolia* subsp. *parvifolia* tuber pulp (%).

Parameter	Mean Value ± SD
Moisture *	88.34 ± 0.35
Dry Matter	11.66 ± 0.35
Ash	9.27 ± 0.77
Protein	3.64 ± 0.10
Fat	3.18 ± 0.25
Crude fiber	15.61 ± 0.44
Carbohydrate	68.27 ± 3.00

The results represent the mean ± standard deviation of the analysis performed in triplicate. * Moisture based on 100 g fresh weight; all other parameters based on 100 g dry weight.

**Table 4 molecules-26-02097-t004:** Total soluble phenols, total soluble flavonoids and antioxidant activity of *Ceiba aesculifolia* subsp. *parvifolia* tuber.

Properties (Wet Sample)	Pulp	Peel
Total soluble phenols (mg GAE/g EE)	44.21 ± 3.48 ^b^	377.99 ± 13.77 ^a^
Total soluble flavonoids (mg CE/g EE)	17.84 ± 1.87 ^b^	134.00 ± 4.47 ^a^
EC_50_ radical DPPH (µg/mL)	12.70 ± 0.13 ^a^	1.22 ± 0.13 ^b^
EC_50_ radical ABTS (µg/mL)	2.47 ± 0.17 ^a^	0.17 ± 0.04 ^b^

Results are expressed as the mean ± standard deviation of the analysis performed in triplicate. Different letters in the same row are significantly different (*p* < 0.05). GAE: gallic acid equivalents; CE: catechin equivalents; EE: ethanolic extract.

**Table 5 molecules-26-02097-t005:** Color parameters and extraction yield of *Ceiba aesculifolia* subsp. *parvifolia* tuber starch.

Parameter	Mean Value ± SD
L*	91.28 ± 0.43
a*	−0.06 ± 0.03
b*	2.45 ± 0.29
Extraction yield (% *w/w*)	23.86 ± 0.40

The results represent the mean ± standard deviation of the analysis performed in triplicate. L* (brightness), C* (chromaticity) and °h (Hue angle).

**Table 6 molecules-26-02097-t006:** Proximate analysis of *Ceiba aesculifolia* subsp. *parvifolia* tuber starch (%).

Parameter	Mean Value ± SD
Moisture *	10.43 ± 0.18
Ash	0.28 ± 0.01
Protein	1.07 ± 0.11
Fat	0.37 ± 0.04
Purity	98.28 ± 0.10

The results represent the mean ± standard deviation of the analysis performed in triplicate. * Moisture based on 100 g fresh weight; all other parameters based on 100 g dry weight.

**Table 7 molecules-26-02097-t007:** Dynamic viscoelastic properties of *Ceiba aesculifolia* subsp. *parvifolia* tuber starch.

Starch Concentration (%)	*G′* (Pa)	*G″* (Pa)	*η** (Pa·s)	Tan(*δ*)
CAP 5	44.64 ^c^	0.34 ^c^	7.08 ^c^	0.13 ^a^
CAP 10	219.34 ^b^	27.18 ^b^	35.03 ^b^	0.12 ^a^
CAP 15	207.13 ^b^	32.03 ^b^	43.70 ^b^	0.16 ^a^
Corn 10 (control)	1503.98 ^a^	70.72 ^a^	238.67 ^a^	0.03 ^b^

Results are the means of two determinations. Different letters in the same column are significantly different (*p* < 0.05). CAP: *Ceiba aesculifolia* subsp. *parvifolia*. *G′*: storage moduli; *G″*: loss moduli; *η**: complex viscosity. Data is presented at a frequency of 1 Hz.

**Table 8 molecules-26-02097-t008:** Power-law model parameters for *Ceiba aesculifolia* subsp. *parvifolia* tuber starch.

Starch Concentration (%)	*K* (Pa·s)	*n* (Adimensional)	R^2^
CAP 5	0.004 ^c^	1.273 ^a^	1.00 ^a^
CAP 10	33.799 ^b^	0.300 ^b^	0.99 ^a^
CAP 15	21.628 ^b^	0.380 ^b^	0.99 ^a^
Corn 10 (control)	122.771 ^a^	0.158 ^b^	0.97 ^b^

Results are the means of two determinations. Different letters in the same column are significantly different (*p* < 0.05). CAP: *Ceiba aesculifolia* subsp. *parvifolia*. *K*: consistency index; *n*: flow behavior index; R^2^: coefficient of determination.

**Table 9 molecules-26-02097-t009:** Thermal and pasting properties of *Ceiba aesculifolia* subsp. *parvifolia* tuber starch.

Thermal Characteristics					Pasting Properties					
**Parameters**	**To (°C)**	**Tp (°C)**	**Tc (°C)**	**ΔH (J·g^−1^)**	**Pv (cP)**	**Tv (cP)**	**Fv (cP)**	**P_temp_ (°C)**	**Breakdown (cP)**	**Setback (cP)**
**Results**	75.12	79.41	84.41	16.82	1556	1029.5	1751.5	65.15	526.5	722

Results are the means of three determinations. To: onset temperature; Tp: peak temperature; Tc: conclusion temperature; ΔH: gelatinization enthalpy; Pv: peak viscosity; Tv: trough viscosity; Fv: final viscosity, P_temp_: pasting temperature.

## Data Availability

Not applicable.

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
