# Peer review of "Physical, Chemical and Rheological Characterization of Tuber and Starch from Ceiba aesculifolia subsp. parvifolia"

_molecules, 2021, doi:10.3390/molecules26072097_

Round 1

Reviewer 1 Report

The manuscript should be impoved before publication. 

The abstract should be rewrite. This sentances is not suitable to be in abstract:
 "hardness (249 N) values were higher than previously reported for another tubers". 

Introduction part: To the best our knowledge, up to the present time the nutritional composition of this vegetable is still little studied. Please check this is three or vegetable. In introduction part your state that it is a tree. Please check and correct. 

Check the statement: "dynamic and steady shear rheological assays". May be "rheological studies" are better as  an expression. 

It is highly recommendable in section 2.4. Characterization of Ceiba aesculifolia subsp. parvifolia Starch to be included the significant data about: 

  1. Yield of starch, %
  2. Purity of the isolated starch.
  3. Measurement of colour (whiteness) of isolated starch.
  4. Molecular mass, coefficient of polysipersity that should explain the rheological and functional properties of isolated starch. 
  5. Protein content in isolated polidsacchride.
  6. In above mentioned analysis there are data for dietary fiber, but infomation about starch content are absent. How the authors descided that this plant is an alternative source of starch? Serious check of these should be done and added as information.
  7. The presented FT-IR spectra showed presence of many impurity, as esters bonds, amino groups, that are not typical for starch. In addition typical bands for alfa glucose is absent and not commented in fingerprint region. Please improve spectra discussion. Repeat the analysis with purified starch. The presnce of impurity strongly influate the rheological properties. Therefore, I did not consider that the demonstrated rheological measurements are due to only to strarch, but may be they interfere with protein or lipid bounded to starch molecule. 
  8. DSC should be better commented. This is improtant for termal stability of starch.

Author Response

Comments and Suggestions for Authors

R1.- The manuscript should be improved before publication. 

The manuscript has been revised and improved

R2.-The abstract should be rewrite. This sentence is not suitable to be in abstract:  "hardness (249 N) values were higher than previously reported for another tubers". 

The sentence “The hardness (249 N) values were higher than previously reported for another tubers” was replaced by “The tuber presented a high hardness value (249 N)”.

R3.- Introduction part: To the best our knowledge, up to the present time the nutritional composition of this vegetable is still little studied. Please check this is three or vegetable. In introduction part your state that it is a tree. Please check and correct. 

The species Ceiba aesculifolia subsp. parvifolia is a tree, however, the tuber is the part of the plant studied in this paper. The term “vegetable” was replaced by “tuber”.

R4.- Check the statement: "dynamic and steady shear rheological assays". May be "rheological studies" are better as an expression. 

The expression has been corrected as indicated by the Reviewer.

R5.- It is highly recommendable in section 2.4. Characterization of Ceiba aesculifolia subsp. parvifolia Starch to be included the significant data about: 

a. Yield of starch, %

Extraction yield was calculated and included in the manuscript.

b. Purity of the isolated starch.

A proximate analysis of the starch was performed to determine the percentages of moisture, ash, lipids and protein. Purity percentage of isolated starch was determined as described by Cruz et al., 2016, using the next equation:

Cruz G, Ribotta P, Ferrero C, Iturriaga L. Physicochemical and rheological characterization of Andean tuber starches: Potato (Solanum tuberosum ssp. Andigenum), Oca (Oxalis tuberosa Molina) and Papalisa (Ullucus tuberosus Caldas). Starch/Staerke. 2016;68(11–12):1084–94. 

https://doi.org/10.1002/star.201600103

c. Measurement of colour (whiteness) of isolated starch.

Color parameters were calculated and included in the manuscript.

d. Molecular mass, coefficient of polydispersity that should explain the rheological and functional properties of isolated starch. 

We agree with the Reviewer suggestion; however, due to the current situation we are unable to carry out these assays. We will take into account subsequently.

e. Protein content in isolated polysaccharide.

Protein content was determined and included in the manuscript.

f. In above mentioned analysis there are data for dietary fiber, but information about starch content is absent. How the authors decided that this plant is an alternative source of starch? Serious check of these should be done and added as information.

Next information was added to the manuscript:

“The extraction yield of CAP tuber starch was of 23.86 %, this is a relatively low value if compared to other tubers such as Solanum tuberosum (32.1 %) (Hoover, 1981) and Colocasia esculenta (81 %) (Agama-Acevedo et al., 2011). Nonetheless, a single root of CAP tree may produce at least four tubers and are deemed that a fully grown tree will have more than 100 tubers, just in their surface roots (Muller, 1952). Although starch is not stored in large quantities in this tuber, the large biomass production of this tree make it a viable source for starch extraction”.

Hoover R, Hadziyev D. Characterization of Potato Starch and Its Monoglyceride Complexes. Starch ‐ Stärke. 1981;33(9):290–300.

          https://doi.org/10.1002/star.19810330903

Agama-Acevedo E, Garcia-Suarez FJ, Gutierrez-Meraz F, Sanchez-Rivera MM, San Martin E, Bello-Pérez LA. Isolation and partial characterization of Mexican taro (Colocasia esculenta L.) starch. Starch/Staerke. 2011;63(3):139–46.

         https://doi.org/10.1002/star.201000113

Muller CH. Los camotes del pochote (Ceiba parvifolia) de Puebla. Bot Sci. 1952;14:18–21.

          https://doi.org/10.17129/botsci.978

g. The presented FT-IR spectra showed presence of many impurity, as esters bonds, amino groups, that are not typical for starch. In addition, typical bands for alfa glucose are absent and not commented in fingerprint region. Please improve spectra discussion. Repeat the analysis with purified starch. The presence of impurity strongly influence the rheological properties. Therefore, I did not consider that the demonstrated rheological measurements are due to only to starch, but may be they interfere with protein or lipid bounded to starch molecule. 

The reviewer is right, there was a mistake in the FTIR figure presented. The FTIR figure was corrected and improved in order to show in detail the fingerprint region. Typical bands for alfa glucose are observed between 575-577 cm-1 bands assigned to pyranose ring; in the same way, the peak observed at 930 cm-1 are due to skeletal mode vibration of α 1→4 glycoside bonds. This concern was not previously highlighted because it was not considered necessary.

The proximal analysis of the starch exhibits the presence of minor compounds such as lipids and protein and this may explain the presence of bands attributed to esters bonds and amino groups.

The percentage of purity (>98%) indicate that starch isolation method results in a starch with low amount of impurities; furthermore, it may be considered as high purity starch (Gao et al., 2009).

Gao J, Vasanthan T, Hoover R. Isolation and characterization of high-purity starch        isolates from regular, waxy, and high-amylose hulless barley grains. Cereal Chem. 2009;86(2):157–63.

h. DSC should be better commented. This is important for thermal stability of starch.

DSC discussion was improved as indicated by the Reviewer

Reviewer 2 Report

The article molecules-1047769 presents a detailed research work that deals with the potential application of Ceiba aesculifolia subsp. parvifolia (CAP) tuber as a thickening agent in food systems requiring a low viscosity and at the same time subjected to high temperatures. The paper is very well written and data are given to readers very clearly. In addition, numerous physicochemical and biochemical analyses have been carried out. Ι have some minor corrections to provide. These follow the text sequence:

-Abstract

Line 24. ''....from the isolated starch...''.

-Introduction

Line 65.Add the word ''the'' instead of''a'' ,i.e.''..the physical, chemical...''.

-Suggestion

Can the authors from their results develop a linear model for predicting rheological, thermal and pasting properties? See for your guidance the study''Journal of Food Engineering, 213, 18-26, 2017''.

Based on the above, I suggest a minor revision of the present article prior to the publication.

Author Response

Comments and Suggestions for Authors

R1.- The manuscript should be improved before publication. 

A1. The manuscript has been revised and improved

R2.-The abstract should be rewrite. This sentence is not suitable to be in abstract:  "hardness (249 N) values were higher than previously reported for another tubers". 

A2. The sentence “The hardness (249 N) values were higher than previously reported for another tubers” was replaced by “The tuber presented a high hardness value (249 N)”.

R3.- Introduction part: To the best our knowledge, up to the present time the nutritional composition of this vegetable is still little studied. Please check this is three or vegetable. In introduction part your state that it is a tree. Please check and correct. 

A3. The species Ceiba aesculifolia subsp. parvifolia is a tree, however, the tuber is the part of the plant studied in this paper. The term “vegetable” was replaced by “tuber”.

R4.- Check the statement: "dynamic and steady shear rheological assays". May be "rheological studies" are better as an expression. 

A4. The expression has been corrected as indicated by the Reviewer.

R5.- It is highly recommendable in section 2.4. Characterization of Ceiba aesculifolia subsp. parvifolia Starch to be included the significant data about: 

1. Yield of starch, %

Extraction yield was calculated and included in the manuscript.

2. Purity of the isolated starch.

A proximate analysis of the starch was performed to determine the percentages of moisture, ash, lipids and protein. Purity percentage of isolated starch was determined as described by Cruz et al., 2016. was calculated and included in the manuscript.

Cruz G, Ribotta P, Ferrero C, Iturriaga L. Physicochemical and rheological characterization of Andean tuber starches: Potato (Solanum tuberosum ssp. Andigenum), Oca (Oxalis tuberosa Molina) and Papalisa (Ullucus tuberosus Caldas). Starch/Staerke. 2016;68(11–12):1084–94. 

https://doi.org/10.1002/star.201600103

3. Measurement of colour (whiteness) of isolated starch.

Color parameters were calculated and included in the manuscript.

4. Molecular mass, coefficient of polydispersity that should explain the rheological and functional properties of isolated starch. 

We agree with the Reviewer suggestion; however, due to the current situation we are unable to carry out these assays. We will take into account subsequently.

5. Protein content in isolated polysaccharide.

Protein content was determined and included in the manuscript.

6. In above mentioned analysis there are data for dietary fiber, but information about starch content is absent. How the authors decided that this plant is an alternative source of starch? Serious check of these should be done and added as information.

Next information was added to the manuscript:

“The extraction yield of CAP tuber starch was of 23.86 %, this is a relatively low value if compared to other tubers such as Solanum tuberosum (32.1 %) (Hoover, 1981) and Colocasia esculenta (81 %) (Agama-Acevedo et al., 2011). Nonetheless, a single root of CAP tree may produce at least four tubers and are deemed that a fully grown tree will have more than 100 tubers, just in their surface roots (Muller, 1952). Although starch is not stored in large quantities in this tuber, the large biomass production of this tree make it a viable source for starch extraction”.

 Hoover R, Hadziyev D. Characterization of Potato Starch and Its Monoglyceride Complexes. Starch ‐ Stärke. 1981;33(9):290–300.

          https://doi.org/10.1002/star.19810330903

Agama-Acevedo E, Garcia-Suarez FJ, Gutierrez-Meraz F, Sanchez-Rivera MM, San Martin E, Bello-Pérez LA. Isolation and partial characterization of Mexican taro (Colocasia esculenta L.) starch. Starch/Staerke. 2011;63(3):139–46.

         https://doi.org/10.1002/star.201000113

Muller CH. Los camotes del pochote (Ceiba parvifolia) de Puebla. Bot Sci. 1952;14:18–21.  https://doi.org/10.17129/botsci.978

7. The presented FT-IR spectra showed presence of many impurity, as esters bonds, amino groups, that are not typical for starch. In addition, typical bands for alfa glucose are absent and not commented in fingerprint region. Please improve spectra discussion. Repeat the analysis with purified starch. The presence of impurity strongly influence the rheological properties. Therefore, I did not consider that the demonstrated rheological measurements are due to only to starch, but may be they interfere with protein or lipid bounded to starch molecule. 

The reviewer is right, there was a mistake in the FTIR figure presented. The FTIR figure was corrected and improved in order to show in detail the fingerprint region. Typical bands for alfa glucose are observed between 575-577 cm-1 bands assigned to pyranose ring; in the same way, the peak observed at 930 cm-1 are due to skeletal mode vibration of α 1→4 glycoside bonds. This concern was not previously highlighted because it was not considered necessary.

The proximal analysis of the starch exhibits the presence of minor compounds such as lipids and protein and this may explain the presence of bands attributed to esters bonds and amino groups.

The percentage of purity (>98%) indicate that starch isolation method results in a starch with low amount of impurities; furthermore, it may be considered as high purity starch (Gao et al., 2009).

Gao J, Vasanthan T, Hoover R. Isolation and characterization of high-purity starch        isolates from regular, waxy, and high-amylose hulless barley grains. Cereal Chem. 2009;86(2):157–63.

8. DSC should be better commented. This is important for thermal stability of starch.

DSC discussion was improved as indicated by the Reviewer

Reviewer 3 Report

Manuscript ID: molecules-1047769

Title: Physical, chemical and rheological characterization of tuber and starch from Ceiba aesculifolia subsp. parvifolia

Journal: molecules

Properties of starch from Ceiba aesculifolia subsp. parvifolia were studied and discussed. Although, the results are interesting, some details should be improved and explained.

I would like to make some comments that authors could take into account to improve the overall quality of the manuscript.

Comments:

  • Line 22: It will be better to write that the average weight of CAP tuber was 3.66 kg (additionally line 72). The information of edible yield (pulp?) should be given before color parameters which describes this part.
  • Line 24 and 25: It is not clear that results based on 100 g dry weight.
  • Paragraf 2.2 and 3.3: It was not mentioned if the results correspond to whole tuber or only pulp?
  • Line 102 and 104: The information is too general, especially if composition of ash was not determined.
  • Table 3: The carbohydrate content in fresh matter is about 8%. Assuming there is starch is dominating carbohydrate, the CAP tuber cannot be recognized as an industrial raw material for starch production. However, authors suggest many times that this starch can be used in food production as raw material.
  • Line 126 and 127: EC50 (µg/mL) was compared with values obtained by other authors but is it justified here? I am not sure that the extraction procedures / weighed amount of samples were the same.
  • Table 4: EC50 or IC50?
  • Lines 208-210: It was written “Ptemp represents the temperature at which viscosity starts increasing, the value obtained for the starch of CAP tuber was higher (79.15 °C)”. The Fig. 3 do not confirm this result, I see that viscosity starts increasing at 65C.
  • Paragraf 3.5.1.: Lack information how samples were pasted (time, temperature)?

Author Response

R1.- The manuscript should be improved before publication.

A1. The manuscript has been revised and improved

R2.-The abstract should be rewrite. This sentence is not suitable to be in abstract: "hardness (249 N) values were higher than previously reported for another tubers".

A2. The sentence “The hardness (249 N) values were higher than previously reported for another tubers” was replaced by “The tuber presented a high hardness value (249 N)”.

R3.- Introduction part: To the best our knowledge, up to the present time the nutritional composition of this vegetable is still little studied. Please check this is three or vegetable. In introduction part your state that it is a tree. Please check and correct.

A3. The species Ceiba aesculifolia subsp. parvifolia is a tree, however, the tuber is the part of the plant studied in this paper. The term “vegetable” was replaced by “tuber”.

R4.- Check the statement: "dynamic and steady shear rheological assays". May be "rheological studies" are better as an expression.

A4. The expression has been corrected as indicated by the Reviewer.

R5.- It is highly recommendable in section 2.4. Characterization of Ceiba aesculifolia subsp. parvifolia Starch to be included the significant data about:

1. Yield of starch, %

Extraction yield was calculated and included in the manuscript.

2. Purity of the isolated starch.

A proximate analysis of the starch was performed to determine the percentages of moisture, ash, lipids and protein. Purity percentage of isolated starch was determined as described by Cruz et al., 2016. was calculated and included in the manuscript.

Cruz G, Ribotta P, Ferrero C, Iturriaga L. Physicochemical and rheological characterization of Andean tuber starches: Potato (Solanum tuberosum ssp. Andigenum), Oca (Oxalis tuberosa Molina) and Papalisa (Ullucus tuberosus Caldas). Starch/Staerke. 2016;68(11–12):1084–94.

https://doi.org/10.1002/star.201600103

3. Measurement of colour (whiteness) of isolated starch.

Color parameters were calculated and included in the manuscript.

4. Molecular mass, coefficient of polydispersity that should explain the rheological and functional properties of isolated starch.

We agree with the Reviewer suggestion; however, due to the current situation we are unable to carry out these assays. We will take into account subsequently.

5. Protein content in isolated polysaccharide.

Protein content was determined and included in the manuscript.

6. In above mentioned analysis there are data for dietary fiber, but information about starch content is absent. How the authors decided that this plant is an alternative source of starch? Serious check of these should be done and added as information.

Next information was added to the manuscript:

“The extraction yield of CAP tuber starch was of 23.86 %, this is a relatively low value if compared to other tubers such as Solanum tuberosum (32.1 %) (Hoover, 1981) and Colocasia esculenta (81 %) (Agama-Acevedo et al., 2011). Nonetheless, a single root of CAP tree may produce at least four tubers and are deemed that a fully grown tree will have more than 100 tubers, just in their surface roots (Muller, 1952). Although starch is not stored in large quantities in this tuber, the large biomass production of this tree make it a viable source for starch extraction”.

Hoover R, Hadziyev D. Characterization of Potato Starch and Its Monoglyceride Complexes. Starch ‐ Stärke. 1981;33(9):290–300.

https://doi.org/10.1002/star.19810330903

Agama-Acevedo E, Garcia-Suarez FJ, Gutierrez-Meraz F, Sanchez-Rivera MM, San Martin E, Bello-Pérez LA. Isolation and partial characterization of Mexican taro (Colocasia esculenta L.) starch. Starch/Staerke. 2011;63(3):139–46.

https://doi.org/10.1002/star.201000113

Muller CH. Los camotes del pochote (Ceiba parvifolia) de Puebla. Bot Sci. 1952;14:18–21. https://doi.org/10.17129/botsci.978

7. The presented FT-IR spectra showed presence of many impurity, as esters bonds, amino groups, that are not typical for starch. In addition, typical bands for alfa glucose are absent and not commented in fingerprint region. Please improve spectra discussion. Repeat the analysis with purified starch. The presence of impurity strongly influence the rheological properties. Therefore, I did not consider that the demonstrated rheological measurements are due to only to starch, but may be they interfere with protein or lipid bounded to starch molecule.

The reviewer is right, there was a mistake in the FTIR figure presented. The FTIR figure was corrected and improved in order to show in detail the fingerprint region. Typical bands for alfa glucose are observed between 575-577 cm-1 bands assigned to pyranose ring; in the same way, the peak observed at 930 cm-1 are due to skeletal mode vibration of α 1→4 glycoside bonds. This concern was not previously highlighted because it was not considered necessary.

The proximal analysis of the starch exhibits the presence of minor compounds such as lipids and protein and this may explain the presence of bands attributed to esters bonds and amino groups.

The percentage of purity (>98%) indicate that starch isolation method results in a starch with low amount of impurities; furthermore, it may be considered as high purity starch (Gao et al., 2009).

Gao J, Vasanthan T, Hoover R. Isolation and characterization of high-purity starch isolates from regular, waxy, and high-amylose hulless barley grains. Cereal Chem. 2009;86(2):157–63.

8. DSC should be better commented. This is important for thermal stability of starch.

DSC discussion was improved as indicated by the Reviewer

Round 2

Reviewer 3 Report

I cannot evaluate a new version of manuscript because the answers to my questions were not attached. I am looking forward your explanations.

My original questions:

Properties of starch from Ceiba aesculifolia subsp. parvifolia were studied and discussed. Although, the results are interesting, some details should be improved and explained.

I would like to make some comments that authors could take into account to improve the overall quality of the manuscript.

Comments:

  • Line 22: It will be better to write that the average weight of CAP tuber was 3.66 kg (additionally line 72). The information of edible yield (pulp?) should be given before color parameters which describes this part.
  • Line 24 and 25: It is not clear that results based on 100 g dry weight.
  • Paragraf 2.2 and 3.3: It was not mentioned if the results correspond to whole tuber or only pulp?
  • Line 102 and 104: The information is too general, especially if composition of ash was not determined.
  • Table 3: The carbohydrate content in fresh matter is about 8%. Assuming there is starch is dominating carbohydrate, the CAP tuber cannot be recognized as an industrial raw material for starch production. However, authors suggest many times that this starch can be used in food production as raw material.
  • Line 126 and 127: EC50 (µg/mL) was compared with values obtained by other authors but is it justified here? I am not sure that the extraction procedures / weighed amount of samples were the same.
  • Table 4: EC50 or IC50?
  • Lines 208-210: It was written “Ptemp represents the temperature at which viscosity starts increasing, the value obtained for the starch of CAP tuber was higher (79.15 °C)”. The Fig. 3 do not confirm this result, I see that viscosity starts increasing at 65C.
  • Paragraf 3.5.1.: Lack information how samples were pasted (time, temperature)?

Author Response

RESPONSES TO REVIEWER 3 COMMENTS
The authors apologize for not sending the response to your comments in the first version of the corrected manuscript.
Here below are the responses to your comments:
1.- Line 22: It will be better to write that the average weight of CAP tuber was 3.66 kg (additionally line 72). The information of edible yield (pulp?) should be given before color parameters which describes this part.
The next paragraph was included in the manuscript in lines 75-78, before color parameters:
“Edible yield of CAP tuber (82.20 %) was similar to the value reported for chayote (Sechium edule) tuber, 86 % [17] and it was higher than the edible yield of sagu (Canna edulis) rhizome, 68.54 % [19]. The weight and percentage of edible yield of CAP tuber suggests that there is scope for its use”.

2.- Line 24 and 25: It is not clear that results based on 100 g dry weight.
The manuscript has been corrected as indicated by the Reviewer. The sentence “reported in dry weight” was added in line 24.

3.- Paragraph 2.2 and 3.3: It was not mentioned if the results correspond to whole tuber or only pulp?
The manuscript has been corrected as indicated by the Reviewer. The results of proximate analysis refer to the pulp.

4.- Line 102 and 104: The information is too general, especially if composition of ash was not determined
The sentence:
“The mineral intake is important to maintain good health, since a deficiency can promote the development or aggravate some illnesses such as hypertension, hyperlipidemia, and coronary heart disease, among others [27].”
Was replaced by:
“The mineral intake is important to maintain a good health, this makes the CAP tuber pulp a potential contributor towards a balanced diet [27].”
The added information is in lines 105-107.

5.- Table 3: The carbohydrate content in fresh matter is about 8%. Assuming there is starch is dominating carbohydrate, the CAP tuber cannot be recognized as an industrial raw material for starch production. However, authors suggest many times that this starch can be used in food production as raw material.
The extraction yield of CAP tuber starch was calculated and the next information was included in the manuscript in lines 151-156:
“The extraction yield of CAP tuber starch was of 23.86 %, this is a relatively low value when compared with other tubers such as Solanum tuberosum (32.1 %) [38] and Colocasia esculenta (81 %) [39]. Nonetheless, a single root of CAP tree may produce at least four tubers and are deemed that a fully grown tree will have more than 100 tubers, just in their surface roots [40]. Although starch is not stored in large quantities in this tuber, the large biomass production of this tree makes it a viable source for starch extraction”.

6.- Line 126 and 127: EC50 (µg/mL) was compared with values obtained by other authors but is it justified here? I am not sure that the extraction procedures / weighed amount of samples were the same.
The sentence:
“The EC50 value in pulp was higher than those reported by Fidrianny et al. [33] for different varieties of Ipomoea batatas, with EC50 values ranged from 45.05 to 98.18 µg/mL by DPPH method”.
Was replaced by:
“The EC50 value in pulp was higher than those reported by Park et al. [33] for yacon (Smallanthus sonchifolius) tuber extract, with an EC50 value of 675.28 µg/mL by DPPH method”.
Park JS, Yang JS, Hwang BY, Yoo BK, Han K. Hypoglycemic effect of Yacon tuber extract and its constituent, chlorogenic acid, in streptozotocin-induced diabetic rats. Biomol Ther. 2009;17(3):256–62.
The added information is in lines 130-132.

7.- Table 4: EC50 or IC50?
The manuscript has been corrected as indicated by the Reviewer. The correct term is EC50

8.- Lines 208-210: It was written “Ptemp represents the temperature at which viscosity starts increasing, the value obtained for the starch of CAP tuber was higher (79.15 °C)”. The Fig. 3 do not confirm this result, I see that viscosity starts increasing at 65C.
The manuscript has been corrected as indicated by the Reviewer. The correct value is 65.15 °C and it was corrected in Table 9.

9.- Paragraf 3.5.1.: Lack information how samples were pasted (time, temperature)?
The next paragraph was included in the manuscript in lines 432-436:
“Starch gels at 5, 10 and 15 % (w/v) were prepared as follows, weighed amounts of starch were mixed with distilled water and immersed in a boiling water bath using a magnetic stirrer equipped with a hot plate. Once a homogeneous dispersion was obtained, the temperature was raised until gelatinization point, which was about 85 °C, and then held at this temperature for 15 min in order to ensure full viscosity development”.
